# Interaction of amphiphilic lipoarabinomannan with host carrier lipoproteins in tuberculosis patients: Implications for blood-based diagnostics

Shailja Jakhar[1], Ramamurthy Sakamuri[1¤], Dung Vu[1,2], Priya Dighe[3], Loreen R. Stromberg[1], Laura Lilley[1], Nicolas Hengartner[4], Basil I. Swanson[3], Emmanuel Moreau[5], Susan E. Dorman[6], Harshini Mukundan[1]*

1 Physical Chemistry and Applied Spectroscopy, Chemistry Division, Los Alamos National Laboratory, Los Alamos, New Mexico, United States of America, 2 Actinide Analytical chemistry, Chemistry Division, Los Alamos National Laboratory, Los Alamos, New Mexico, United States of America, 3 Biosecurity and Public Health, Bioscience Division, Los Alamos National Laboratory, Los Alamos, New Mexico, United States of America, 4 Theoretical Biology and Biophysics, Theory Division, Los Alamos National Laboratory, Los Alamos, New Mexico, United States of America, 5 Foundation for Innovative New Diagnostics, Geneva, Switzerland, 6 Department of Medicine, Medical University of South Carolina, Charleston, South Carolina, United States of America

¤ Current address: Bako Diagnostics, Alpharetta, Georgia, United States of America
* harshini@lanl.gov

**Data Availability Statement:** All relevant data are within the manuscript and its Supporting information files.

## Abstract

Lipoarabinomannan (LAM), an amphiphilic lipoglycan of the *Mycobacterium tuberculosis* cell wall, is a diagnostic target for tuberculosis. Previous work from our laboratory and others suggests that LAM is associated with host serum lipoproteins, which may in turn have implications for diagnostic assays. Our team has developed two serum assays for amphiphile detection: lipoprotein capture and membrane insertion. The lipoprotein capture assay relies on capture of the host lipoproteins, exploiting the biological association of host lipoprotein with microbial amphiphilic biomarkers to "concentrate" LAM. In contrast, the membrane insertion assay is independent of the association between pathogen amphiphiles and host lipoprotein association, and directly captures LAM based on its thermodynamic propensity for association with a supported lipid membrane, which forms the functional surface of an optical biosensor. In this manuscript, we explored the use of these assays for the detection of LAM in sera from adults whose tuberculosis status had been well-characterized using conventional microbiological tests, and endemic controls. Using the lipoprotein capture assay, LAM signal/noise ratios were >1.0 in 29/35 (83%) individuals with culture-confirmed active tuberculosis, 8/13 (62%) individuals with tuberculosis symptoms, but no positive culture for *M. tuberculosis*, and 0/6 (0%) symptom-free endemic controls. To evaluate serum LAM levels without bias associated with potential differences in circulating host lipoprotein concentrations between individuals, we subsequently processed available samples to liberate LAM from associated host lipoprotein assemblies followed by direct detection of the pathogen biomarker using the membrane insertion approach. Using the membrane insertion assay, signal/noise for detection of serum LAM was greater than that observed using the

**Funding:** This work was supported by Los Alamos LDRD Directed Research Grant (Co-PI Mukundan) and grant R21 AI03599 to Dr. Susan Dorman from the National Institutes of Health. Los Alamos National Laboratory, an affirmative action equal opportunity employer, is managed by Triad National Security, LLC for the U.S. Department of Energy's NNSA, under contract 89233218CNA000001. The funder provided support in the form of salaries for authors [HM, SJ and RMS from LDRD; SD and HM from NIAID], but did not have any additional role in the study design, data collection and analysis, decision to publish, or preparation of the manuscript. The specific roles of authors are articulated in the 'author contributions' section.

**Competing interests:** Scientists from the Los Alamos National Laboratories, operated by the Triad LLC, that are authors on this manuscript, do not have competing interests, and are not consultants for any competing interests. Ramamurthy Sakamuri was a post-doctoral researcher at Los Alamos National Laboratory when he conducted this research. He has subsequently transitioned to Bako Diagnostics, which is his current affiliation. However, Bako Diagnostics did not participate or were not involved in the research presented in this manuscript in any way. This does not alter our adherence to PLOS ONE policies on sharing data and materials.

lipoprotein capture method for culture-confirmed TB patients (6/6), yet remained negative for controls (2/2). Taken together, these results suggest that detection of serum LAM is a promising TB diagnostic approach, but that further work is required to optimize assay performance and to decipher the implications of LAM/host lipoprotein associations for diagnostic assay performance and TB pathogenesis.

## Introduction

Tuberculosis (TB) is the leading cause of global mortality associated with a single infectious disease, and is estimated to afflict 10 million people worldwide (2018), with ~ 1.3 million deaths [1]. The World Health Organization has identified the need for a non-sputum diagnostic test for TB, particularly extrapulmonary TB and pulmonary TB associated with low bacillary burden in airways, as can occur in young children and in individuals with HIV co-infection [2].

Accordingly, several biomarkers have been explored for the empirical diagnosis of TB, with lipoarabinomannan (LAM) arguably being the most studied [3–5]. LAM is an amphiphilic lipoglycan component of the *Mycobacterium tuberculosis* (MTB) cell wall that has in vitro immunomodulatory activity including activation of the Toll-like receptor 2 pathway [6–9]. Following the findings of Hamasur *et al.* that LAM was detectable in mouse urine within one day after intra-peritoneal injection of crude MTB cell wall extract, most clinical diagnostic work focused on detection of LAM in urine [10–13]. One lateral flow urinary LAM assay is now commercially available (Alere Determine™ TB LAM Ag, Abbott Biotechnologies). However, the sensitivity of the Alere assay is suboptimal—ranging from 42% in HIV-negative TB patients to 53% in TB patients with advanced HIV disease, a condition in which total mycobacterial burden can be very high and occult renal TB can be present [6, 14, 15]. The next generation Fujifilm SILVAMP TB-LAM (FujiLAM; Fujifilm, Tokyo, Japan), a lateral flow test incorporating high-affinity monoclonal anti-LAM antibodies, has 30% better sensitivity compared to Alere LAM but needs further validation in clinical settings [16]. Several other LAM assay formats including FujiLAM with enhanced sensitivity are in development [11, 16].

The amphiphilic biochemistry of LAM confers instability in aqueous milieu such as blood. Previous work from our team has shown that, in human blood, LAM associates with host lipoproteins such as high-density lipoproteins (HDL). In aqueous blood, HDL is a stable lipidic assembly comprised of a core lipid nanodisc stabilized by coat apolipoproteins [15–17]. While LAM has been extracted from blood of TB patients [9], direct measurement of LAM in blood or serum has proved to be more elusive, and achieved mainly in individuals with advanced HIV disease [9, 18, 19]. We hypothesized that sequestration of LAM in host lipoprotein assemblies may contribute to the difficulty in detecting the antigen in blood. In parallel assessment of LAM in serum and urine from TB patients using an electrochemiluminescence immunoassay, Broger *et al.* showed substantially lower assay sensitivity in serum than in urine, but that matrix inhibition of serum could largely be reversed by heat treatment, resulting in substantial increases in LAM signal in tested sera [20].

To evaluate the impact of serum sequestration of LAM in host lipoprotein complexes, we measured serum LAM using two methods tailored for the detection of amphiphilic biomarkers in aqueous matrices (Fig 1)–lipoprotein capture and membrane insertion. The lipoprotein capture assay (Fig 1) relies on capture of host lipoproteins, exploiting their biological association with the pathogen amphiphile to "concentrate" LAM [8, 21]. In contrast, the membrane insertion assay (Fig 1) is independent of that host lipoprotein/LAM association, and directly

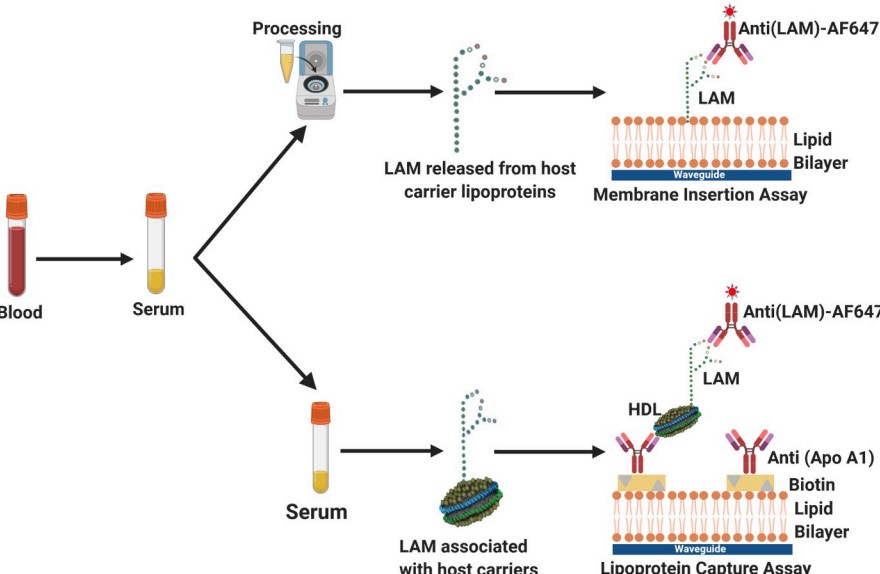

**Fig 1. Overview of lipoarabinomannan (LAM) detection strategies.** When LAM is associated with a host lipoprotein carrier such as HDL, detection can be performed using lipoprotein capture, which requires two antibodies, as well as prior knowledge of LAM-lipoprotein carrier associations. An antibody targeting apolipoprotein A1, the coat protein of HDL, is used to capture the nanodiscs on the assay surface, followed by detection with a fluorescently labeled antibody targeting LAM. In the absence of sequestration by a host lipoprotein carrier, LAM can be directly detected by membrane insertion, which requires only one antibody. The amphiphilic antigen, LAM, is allowed to partition into a supported lipid bilayer interface, followed by detection with a specific fluorescently labeled antibody. Graphic representations are not drawn to scale. Figure created with BioRender.com.

captures LAM based on its thermodynamic propensity for association with a supported lipid membrane which forms the functional surface of a biosensor [19, 21, 22]. Although both of these assays are platform ambivalent, we used enzyme-linked immunosorbent assays (ELISA) and fluorescence measurements from a waveguide-based biosensor platform developed at the Los Alamos National Laboratory for this study [8, 23]. There are two limitations that impact the use of conventional plate-based methods for the detection of LAM. 1) LAM is an amphiphile and therefore, sticks to plastics and other surfaces, reducing assay reliability, sensitivity and reproducibility. 2) In clinical samples, LAM is already sequestered by host lipoproteins and not available in free-form [8]. It must be detected in this conformation or liberated to facilitate detection. In this manuscript, we have attempted to develop methods that address both these limitations. Both the assays are performed on a surface functionalized with lipid bilayer, and therefore override the challenges of using plastic ELISA plates in concert with amphiphiles. Further, we have consistently shown that the use of the waveguide platform offers at least 10x greater sensitivity than conventional plate-based ELISA, which is an advantage when it comes to sensitive detection of pathogen antigens in complex clinical samples [17, 24].

In this manuscript, we evaluated the use of the above two assays- lipoprotein capture and membrane insertion- for the direct detection of LAM in serum from carefully characterized samples from tuberculosis patients, and endemic controls.

## Methods

### Clinical specimens

This study used existing stored specimens that previously had been obtained from participants in Uganda for a study that evaluated the diagnostic accuracy of the Alere Determine™ TB LAM

Ag assay [25]. That diagnostic accuracy study enrolled HIV-positive adults suspected of having active tuberculosis based on the presence of at least one of cough, fever, night sweats, or weight loss. Individuals were excluded if they had received more than two days of anti-tuberculosis treatment. At enrollment, each participant provided two sputum specimens, each of which was cultured in liquid and solid media. One mycobacterial blood culture, performed using the Myco/F LYTIC system (Becton and Dickinson, Franklin Lakes, NJ), was performed for each participant at enrollment. A participant was considered to have active TB if *M. tuberculosis* was isolated in culture from any specimen. Neither Xpert MTB/RIF nor other nucleic acid amplification test was performed on sputum, since those tests were not available on-site at the time of study enrollment. At enrollment, blood was drawn into a BD Vacutainer serum separator tube (Becton and Dickinson), and serum was subsequently withdrawn and immediately frozen at -80 ˚C until used for this study. For this exploratory study, one of the investigators (S.E.D.) selected specimens based on knowledge of participant microbiological classification, with intent to include a representative spectrum of participants with and without culture-confirmed TB. Samples were selected on the basis of culture results and sent to Los Alamos National Laboratory in a double blinded fashion. S2 Table in S1 File describes the demographic information specifically, gender, age, CD4 cells/mm$^3$, identification of infiltrates and miliary infiltrates, cavity and urine LAM diagnostics (where available) for these samples. Additionally, we used existing de-identified serum specimens from six Ugandan adults who did not have TB symptoms and were not known to be HIV-positive; no additional meta-data were available for these specimens. Samples were thawed immediately prior to use for the studies described here. If multiple assays were performed on a single serum sample, lipoprotein capture was performed first with the fewest possible freeze/thaw cycles to avoid degradation of lipoprotein carriers.

## Ethics

This study was approved by ethics committees of Johns Hopkins University School of Medicine, the Joint Clinical Research Centre (Kampala, Uganda), and Los Alamos National Laboratory. All participants provided written informed consent.

## Reagents and materials

Anti-LAM monoclonal antibody (CS40), rabbit anti-LAM polyclonal antibody, and purified LAM (H37Rv) used in validation and optimization assays were obtained from Biodefense and Emerging Infections Resources (BEI resources, Manassas, VA). Anti-LAM monoclonal antibodies used in the reporter cocktail (see below) were a generous gift from the Foundation of Innovative New Diagnostics (FIND, Geneva, Switzerland). Biotinylated anti-ApoA1 antibody (ab27630) was purchased from Abcam (Cambridge, MA). Alexa Fluor 647 conjugated streptavidin (S21374), 1-Step Ultra TMB-ELISA Substrate Solution (34028), EZ-Link Plus Activated Peroxidase kits, Alexa Fluor 647 labelling kits, and polystyrene flat-bottom 96 well plates (Corning 9017) were purchased from Thermo Fisher Scientific (Waltham, MA). Bovine serum albumin (BSA, A7906) and Dulbecco's phosphate buffered saline (PBS, D1408) were obtained from Sigma Aldrich (St. Louis, MO). Human serum was obtained from Fischer Scientific Inc (Catalogue. No. BP2657100). 1, 2-Dioleoyl- sn-glycero-3-phosphocholine (DOPC) and 1, 2-dioleoyl-sn-glycero-3-phosphoethanolamine-N- (cap biotinyl) (sodium salt) (cap Biotin) were obtained from Avanti Polar Lipids (Alabaster, AL).

## Waveguide-based optical biosensor

The waveguide-based optical biosensor was developed at Los Alamos National Laboratory and is described in detail elsewhere [23]. Waveguides were custom engineered by nGimat Inc.

(Norcross, GA) and the surface chemistry was performed at Spectrum Thin Films (Hauppauge, NY). Silicone gaskets for waveguide assembly were from Grace Bio-Labs (Bend, OR) and Secure seal spacers (9 mm diameter x 0.12 mm deep) were from Electron Microscopy Sciences (Hatfield, PA). Glass microscope slides used as coverslips were purchased from Thermo Fisher Scientific (Rockford, IL).

## Waveguide preparation and flow cell assembly

Single mode planar optical waveguides were used for functionalization as previously described [26]. Briefly, waveguides and glass coverslips were cleaned by sequential sonication in chloroform, ethanol and water (5 min each), followed by drying under argon stream and exposure to UV-ozone (UVOCS Inc., Montgomeryville, PA) for 40 min. Flow cells for immunoassays were assembled using clean waveguides and cover slips, which were bonded together with a silicone gasket containing a laser cut channel creating a flow cell. Following assembly, the flow cell was injected with 70 μL of lipid micelles (preparation described below) and then incubated overnight at 4 ˚C to facilitate vesicle fusion and lipid bilayer stabilization.

## Lipid micelle preparation

1, 2-Dioleoyl-sn-glycero-3-phosphocholine (DOPC) and 1, 2-dioleoyl-sn-glycero-3-phosphoethanolamine-N- (cap biotinyl) (sodium salt) (cap biotin) were obtained from Avanti Polar Lipids (Alabaster, AL), resuspended in chloroform and stored at −20 ˚C. Lipid micelles for use in waveguide experiments were prepared as described previously [19]. Briefly, 2 mM DOPC and 1% cap biotinyl (mol/mol) were combined in a glass tube then the chloroform was evaporated off under argon gas. Lipids were rehydrated in PBS, incubated in the dark for 30 min at room temperature with shaking (100 rpm) on an orbital shaker. Lipid solutions then underwent 10 rapid freeze/thaw cycles alternating between liquid nitrogen and room temperature water. Finally, lipids were probe sonicated for 6 min total (1.0 sec pulse on/off, 10% amplitude) using a Branson ultrasonic generator. Once the lipids were stabilized, the addition of biotin allowed for the bilayer integrity to be evaluated during immunoassay experiments by probing with 50–100 pM of a streptavidin Alexa Fluor 647 conjugate [22, 26].

## Waveguide-based assays

All incubations occurred at room temperature. Dilutions of all reagents were made in PBS. Flow cells were prepared as described above and the lipid bilayer was blocked for 1 hr with 2% BSA in PBS (w/v). All incubations were immediately followed by a wash with 2 mL of 0.5% BSA in PBS (w/v) to remove any unbound constituents. Incident light from a 635 nm laser (Diode Laser, Coherent, Auburn, CA) with power adjusted to 440–443 μW was coupled into the waveguide using a diffraction grating. The response signal was adjusted for maximum peak intensity using a spectrometer (USB2000, Ocean Optics, Winter Park, FL) interfaced with the instrument and an optical power meter (Thor Labs, Newton, NJ) [27].

The background signal associated with the lipid bilayer and 2% BSA block was recorded, and then the integrity of the lipid bilayer was assessed by incubation of 50–100 pM streptavidin, AF647 conjugate (Molecular Probes, S32357) for 5 min. The two control steps are performed in every experiment as intrinsic controls. The remaining assay steps depended on the particular assay as described below. The antibodies used in this assay (FIND Clones 171 and 24) were labeled with AF647, and the optimal combination of antibodies and their concentrations were determined using enzyme-linked immunosorbent assays (S1 Fig, S1 Section in S1 File). The incubation times for the assays were optimized in all cases by standard

measurements using LAM spiked into commercially procured human serum. The antigen titrations were performed on the waveguide-based biosensor.

**Lipoprotein capture assay.** Host HDL lipoproteins are nanodiscs of lipids that are held together by a coat protein, Apolipoprotein A1. The lipoprotein capture assay utilized an anti-apoA1 capture antibody for the capture of HDL lipoproteins onto the sensing surface. Following the test for lipid bilayer integrity (instrument controls), 10 nM unlabeled streptavidin was added and incubated for 10 min to saturate the biotin embedded in the lipid bilayer. Next, 100 nM of biotin conjugated α-apoA1 (α-HDL) antibody was added and incubated for 45 min, allowing for the capture antibody to adhere to the surface via biotin-streptavidin interaction. The surface is now functionalized with the capture antibodies for the lipoprotein capture assay. Prior to experimental measurement, however, the non-specific signal was determined by incubation of the fluorescence reporter antibody, FIND antibody cocktail labeled with AF647 (15 nM each antibody, for 45 min), with control human serum on to the waveguide surface. This allows for the determination of the fluorescence signal associated with the interaction of the reporter antibody with the surface and control serum, in the absence of the antigen (no-antigen control).

Upon completion of the control measurements above, the antigen was added, and specific interaction between LAM and the reporter antibody cocktail was measured. To generate standard LAM concentration curves, increasing concentrations of MTB H37RV LAM (100, 250, 500, 1000, 1500, 2500, 5000 nM) were spiked into commercially procured human serum, and incubated for 24 hours to allow for complete association with lipoproteins. For clinical specimens, 200 μL of serum was used for each assay, and directly added to the flow cell. Upon incubation, the FIND reporter cocktail was again added, and the specific signal associated with the binding of LAM with the antibodies was measured via the spectrometer interface.

Three sets of controls were performed (n = 25 each). The instrument background signal is an assessment of the biosensor function and bilayer integrity. No antigen control experiments were performed using control serum, in the absence of LAM. Specificity controls were performed using anti-LAM antibodies (FIND 171 and 24), and measuring the signal associated with their interactions with LAM functionalized on the biosensor surface. In all experiments, raw data were recorded as relative fluorescence units (RFU) as a function of wavelength (nm). The specific/non-specific ratio (S/N) was determined by taking the maximum RFU value for the specific signal, subtracting out the RFU value for the instrument controls, specificity and no-antigen controls (henceforth referred to as the background) and dividing this by the maximum RFU value for the non-specific signal minus the maximum RFU value for the background [Eq (1)].

$$S/N = \frac{(Specific - Background)}{(NS - Background)} \tag{1}$$

**Membrane insertion assay.** For this assay, LAM is released from host lipoprotein complexes prior to detection via a pre-established sample processing method [28]. Briefly, processing was performed using a modified single-phase Bligh and Dyer chloroform:methanol extraction. Chloroform, methanol and LAM sample (either standard or clinical) were combined in a siliconized microfuge tube (Fisher Scientific, 02-681-320) at a 1:2:0.8 (v/v) ratio. The chloroform, methanol, and serum mixture was combined by gentle pipetting using low-retention pipet tips to avoid lipid adherence to the plastic, and then the mixture was centrifuged for 1 min at 2,000 x g to separate the proteins (supernatant) from the lipid/amphiphilic molecules (pellet). The supernatant was discarded and the LAM-containing pellet was resuspended in PBS by gentle pipetting. Following a 5 sec pulse spin to settle debris that could clog the septum of the biosensor flow cell, the LAM-containing solution was used as the biomarker sample for immunoassays.

There is no need for a capture antibody (reduces assay time by 40 min.) in the membrane insertion assay format, as it relies on the direct interaction of the LAM antigen (liberated from carrier assemblies as described above) into the supported bilayer interface. To generate standard concentration curves, LAM antigen was diluted to the desired concentration in control human serum in high-recovery glass vials (Thermo Scientific, Waltham, MA) C5000-995 and incubated overnight (18–24 hrs) at 4 ˚C to allow for association with lipoproteins in serum, as described above the lipoprotein capture assay. The samples were serially diluted, as described above for the lipoprotein capture assay. Each dilution was then subject to the sample processing method, and evaluated in the assay format in order to generate the standard curve. For the clinical samples, 50 µL of each serum sample from patients and controls was subjected to the sample processing method, and used in the assay.

For this set of assays, the three control measurements described above (instrument controls, no-antigen control and specificity controls) were performed (n = 10 each) as well, and the concentration of the reporter antibodies was the same as used for the lipoprotein capture assay above (15 nM for 45 min.). For the experimental measurements, 200 µL of the processed sample was incubated in the flow cell, allowing for association of amphiphilic biomarkers with the lipid bilayer. Then the FIND antibody cocktail was added again, and incubated (15 nM for 45 min) for assessment of the specific signal. Raw data were recorded as relative fluorescence units (RFU) as a function of wavelength (nm).

For both the lipoprotein capture assay and the membrane insertion assay, a S/N ratio > 1.0 was considered a positive result, and each sample measurement was repeated two times in order to assess reproducibility. The laboratory team performing LAM assays using participant specimens was blinded to participant group assignment and other clinical information, and it was held by one of the team members (S.E.D.) as described earlier.

For use of waveguide-based sensor at the point of need, our team has simplified the waveguide-based optical biosensor for field deployment, and validated assay performance on this new portable (<10 lbs) instrument with optical components integrated and the software for functionality on a phone-based application. For easier use of sample processing method, we have developed and validated the first microfluidics device which is capable of performing lipid extractions within minutes at the point of need.

## Statistical analysis

S/N ratios are presented as means ± standard deviation. Welch's t-test and Mann Whitney U-test was used to determine statistical significance. A significance level (P) of less than 0.05 was considered statistically significant (***$P < 0.001$, **$P < 0.01$, or *$P < 0.05$). Outlier analysis was performed using Chauvenet's criterion, which identifies the probability that a given data point reasonably contains all samples in a data set. LAM concentration curve and all significance tests were performed using GraphPad Prism 8.

**Limit of detection.** The limit of detection (LOD) was obtained as described in Eq (2). For a given sample concentration, the average non-specific signal for all replicates was obtained and added to three times the standard deviation ($\sigma$), multiplied by the sample concentration, and divided by the average specific signal for that concentration. Sample concentration and LOD will be in the same units, so if sample concentration is in nM then LOD will be in nM.

$$\text{LOD} = \frac{(NS + 3\sigma)[\text{Sample}]}{Specific} \qquad (2)$$

## Results

### Antibody selection and optimization

For both the lipoprotein capture assay and the membrane insertion assay, antibodies were selected and concentrations determined by enzyme-linkedimmunosorbent assays (ELISAs) (S1 Fig, S1 Section in S1 File). Briefly, antibodies were chosen based on sensitivity and specificity for LAM detection, and a combination of two different monoclonal antibody clones (24 and 171) yielded best outcomes for LAM detection in serum samples. These two antibodies were used as a cocktail at 15 nM each for both the membrane insertion and lipoprotein assay formats.

### Optimization of the lipoprotein capture assay

Fig 2a shows a representative spectral measurement on the waveguide-based biosensor [19, 23] for the measurement of LAM (1.5 μM) spiked and incubated overnight in control human serum. RFU is plotted as a function of emission wavelength (nm), as measured on the spectrometer interface associated with the instrument. LAM concentration curve (Fig 2b) shows a sigmoidal fit with a $R^2$ value of 0.999 and limit of detection- 69 nM, which corresponds to a concentration of 1900 ng/mL LAM.

### Optimization of the membrane insertion assay

Fig 3a shows a representative spectral measurement for LAM (0.5 μM, RFU), using the membrane insertion assay following extraction from spiked serum. The LAM concentration curve (Fig 3b) using this method shows a sigmoidal fit with a $R^2$ value of 0.998 and limit of detection- 3.3 nM, which corresponds to a concentration of 114 ng/mL.

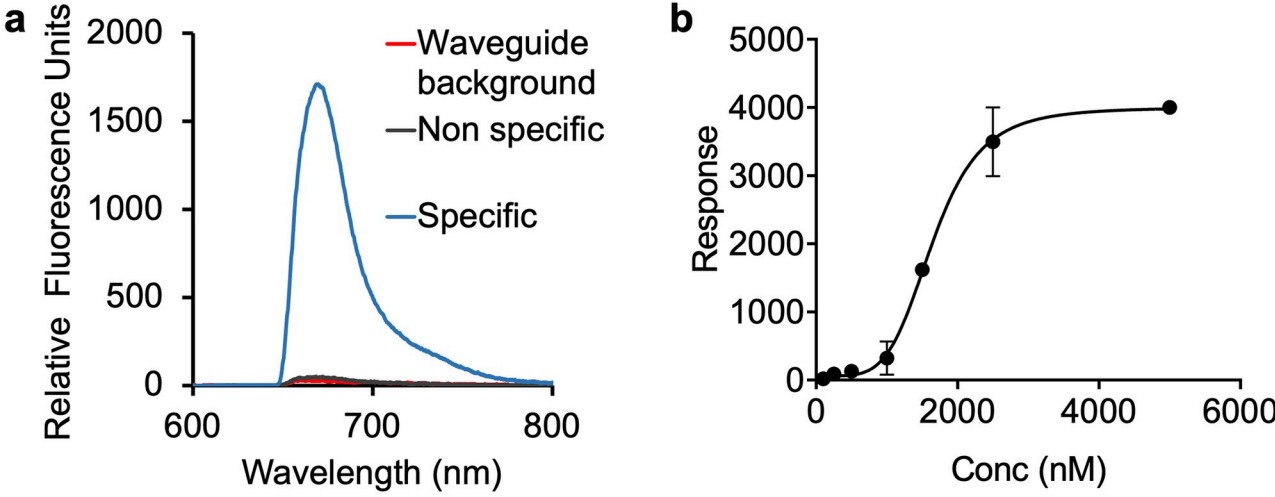

**Fig 2. Assay optimization for the detection of LAM in human serum by lipoprotein capture assay.** Measurement of LAM by lipoprotein capture assay, as a function of concentration. **(a)** Representative spectral measurement of LAM (1.5 μM) incubated overnight at 4 ˚C in control human serum, with the specific signal (relative fluorescence units, RFU) from the detection of α-LAM antibody (15 nM) as a function of emission wavelength (nm). The background and non-specific signals are measured before the addition of LAM. **(b)** Lipoprotein capture assay was performed for the detection of LAM spiked into control serum at various concentrations and incubated overnight to allow incorporation of the amphiphile into carrier assemblies. Results are plotted as RFU as measured on the waveguide-based optical biosensor, at increasing concentrations of LAM. All values given in **(b)** are the mean ± standard deviation derived from at least two independent determinations (n = 2). Statistical significance was determined by Welch's t-test using GraphPad Prism 8.

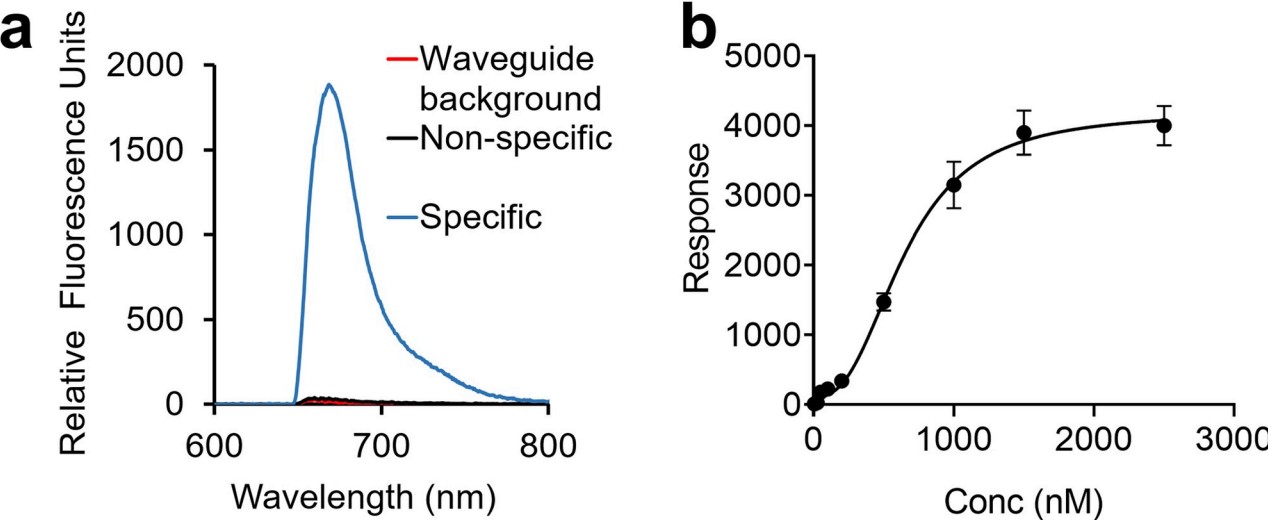

**Fig 3. Assay optimization for the detection of LAM in human serum by membrane insertion assay.** Measurement of LAM by membrane insertion assay, as a function of concentration. **(a)** Representative spectral measurement of LAM (0.5 μM) incubated overnight at 4 ˚C in control human serum, with the specific signal (relative fluorescence units, RFU) from detection of α-LAM antibody (15 nM) as a function of emission wavelength (nm). The background and non-specific signals are measured before the addition of LAM. **(b)** Membrane insertion assay was performed for the detection of LAM spiked into control serum at various concentrations and incubated overnight to allow incorporation of the amphiphile into carrier assemblies. Sample processing was done to remove lipoproteins. Results are plotted as RFU as measured on the waveguide-based optical biosensor, at increasing concentrations of LAM. All values given in **(b)** are the mean ± standard deviation derived from at least two independent determinations (n = 2). Statistical significance was determined by Welch's t-test using GraphPad Prism 8.

### Detection of LAM in clinical samples

Using the lipoprotein capture assay, LAM signal/noise ratio (S/N) was > 1.0 in 29/35 (83%) culture-confirmed TB patients, 8/13 (62%) individuals with TB symptoms but no positive cultures, and 0/6 (0%) healthy controls (Fig 4). Mean S/N ± SD values were 3.8 ± 4.7, 1.9 ± 1.4, and 0.6 ±.20, respectively (Table 1).

To further understand the LAM lipoprotein capture assay performance, we stratified culture-confirmed TB patients by specimen source (sputum and/or blood) of positive MTB culture(s). Surprisingly, there was no association between MTB detected in blood culture, and LAM detected in serum. Serum LAM S/N was >1.0 in 10/12 (83%) culture-confirmed TB patients with MTB in blood cultures vs. 19/23 (83%) culture-confirmed TB patients whose blood culture was negative for MTB (relative risk 1.01, 95% CI 0.68, 1.28). Median (IQR) LAM S/N was 2.2 vs 1.3 among culture-confirmed TB patients with vs. without MTB in blood cultures (Table 1).

We hypothesized that, if host HDL concentration impacted the outcome of the lipoprotein capture assay, then use of a LAM assay approach that was independent of host lipoproteins—membrane insertion- might increase assay analytical sensitivity. Fig 5a shows the comparison of the two methods for detection of LAM in a serum sample spiked with 500 nM of LAM, with all other parameters held constant. Specific signal was significantly greater (10x) using the membrane insertion assay as compared to the lipoprotein capture assay (p = 0.04, $R^2$ = 0.99).

Subsequently we performed the membrane insertion and lipoprotein capture assay in parallel for the same eight clinical samples with sufficient volume for comparative testing (Fig 5b). For serum from culture-confirmed TB patients, the S/N was uniformly higher for the membrane insertion assay than for the lipoprotein capture assay; no specific signal was detected in healthy control sera by either assay.

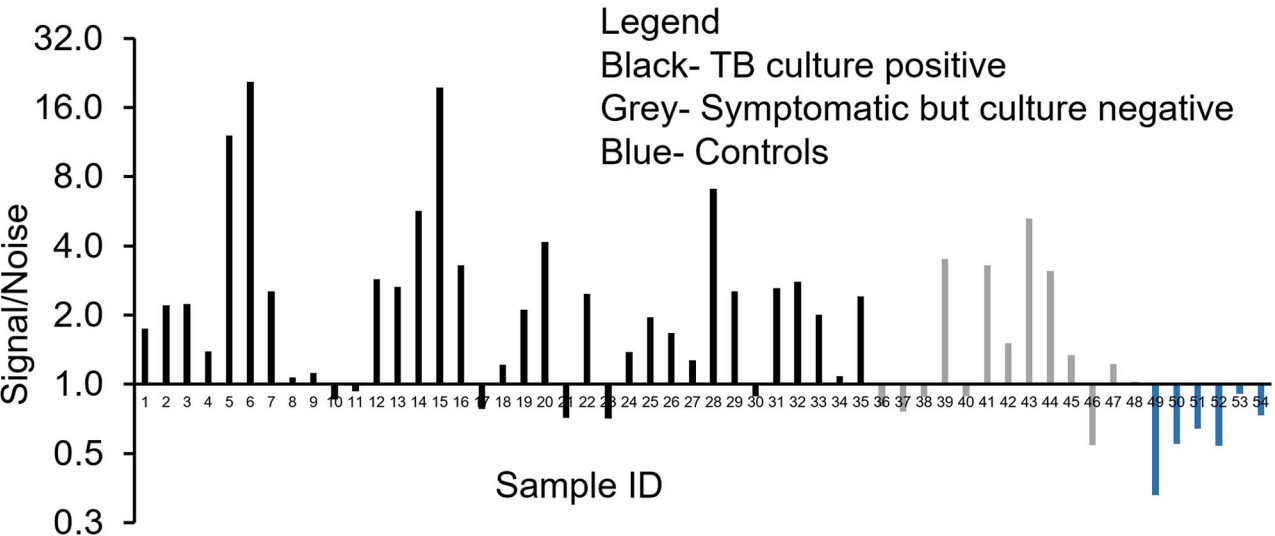

**Fig 4. Direct detection of LAM in patient serum samples.** Detection of LAM in clinical serum samples using the lipoprotein capture assay. Data are presented as the Signal/Noise (S/N) ratio with a value above 1.0 indicating a positive result. The measured S/N in sera from 54 patients from 3 different categories (see legend) is plotted. Results of urine testing using the Determine™ TB LAM Ag assay at the time of enrollment into the clinical study: patients 1–20 were positive, patients 21–48 were negative, and healthy controls 49–54 were not tested.

## Discussion

In this exploratory study, we compared and contrasted the use of two tailored methods for the detection of amphiphilic biomarkers in aqueous samples—lipoprotein capture and membrane insertion—for the measurement of serum LAM.

Both these methods were able to directly measure LAM in serum, with a demonstrated enhancement of sensitivity using the membrane insertion method.

In our initial evaluation in serum from adults whose TB status had been rigorously characterized by conventional mycobacteriology testing, we observed a clear difference between culture-confirmed TB cases and adult controls with regard to both proportion with detectable LAM signal and LAM S/Ns. This finding demonstrates the applicability of these two tailored methods for serum amphiphilic LAM detection.

Clinically, there were two unexpected findings. Our working hypothesis—that serum LAM was associated with presence of MTB in blood cultures—was not supported by the lipoprotein capture assay data, as serum LAM was detected in the majority of culture-confirmed TB

**Table 1. Signal to noise ratios, by clinical group, for the lipoprotein capture assay.**

| Clinical Group | % with SNR >1.0 (n/n) | S/N Median (IQR) | S/N Mean (SD) |
|---|---|---|---|
| **Culture-confirmed TB (HIV-positive)** | **83 (29/35)** | **2.2 (2.6)** | **3.8 (4.7)** |
| Sputum culture-MTB, blood culture MTB | 100 (9/9) | 2.2 (6.2) | 5.0 (6.4) |
| Sputum culture negative, blood culture MTB | 33 (1/3) | 0.9 (5.7) | 2.8 (2.7) |
| Sputum culture MTB, blood culture negative | 83 (19/23) | 2.5 (2.5) | 3.5 (4.0) |
| **TB symptoms but all cultures negative** | **62 (8/13)** | **1.3 (0.4)** | **1.9 (1.4)** |
| **Controls** | **0 (0/6)** | **0.6 (2.3)** | **0.6 (.20)** |

Abbreviations: TB, tuberculosis; MTB, *Mycobacterium tuberculosis*; S/N, signal to noise ratio; SNR, signal to noise ratio; IQR, inter quartile range; SD, standard deviation; RFU, relative fluorescence units.

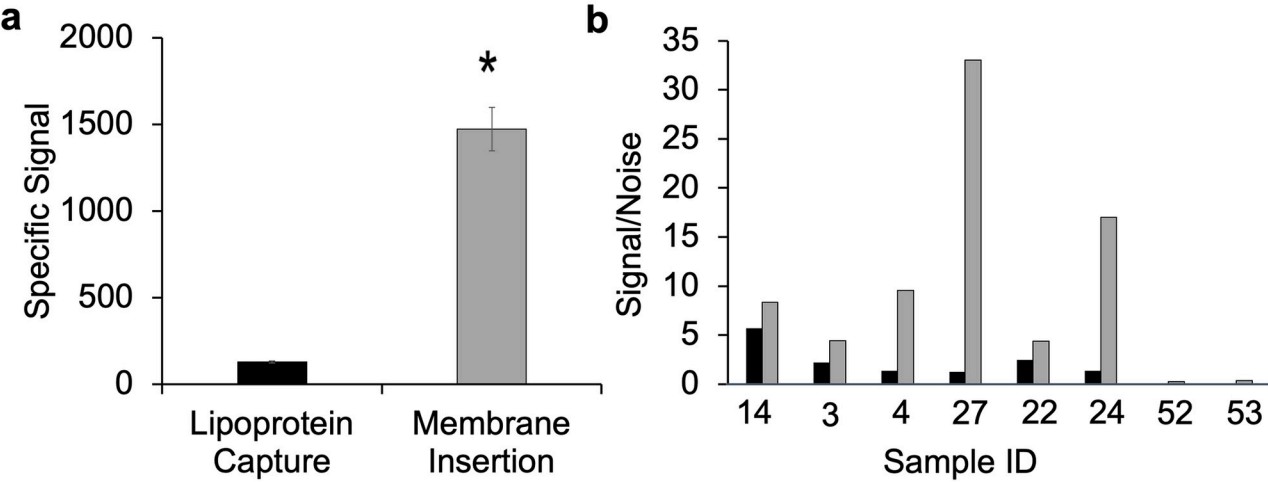

**Fig 5. Comparison of lipoprotein capture and membrane insertion. (a)** Representative measurement of LAM (0.5 μM), by lipoprotein capture (black bars) and membrane insertion assay (grey bars), incubated overnight at 4 ˚C in control human serum, with the specific signal (RFU) from the detection of α-LAM antibody (15 nM). Values are the mean ± standard deviation derived from at least two independent determinations (n = 2). Statistical significance was determined by Welch's t-test (*$P < 0.05$). **(b)** Comparison of LAM detection signal by lipoprotein capture (black bars) and membrane insertion assay (grey bars) in patient serum samples. Data are presented as the S/N ratio with a value over 1 indicating a positive result. Samples 14, 3, 4, 27, 22 and 24 are positive for LAM by either blood or sputum culture methods, whereas samples 52, 53 are healthy controls.

patients whose blood cultures were negative for MTB, and further, was not detected in the few patients whose blood cultures were positive for MTB. This outcome can be either because of an absence of serum LAM, or simply be associated with a failure to pull-down host lipoprotein/ failure to detect LAM due to LAM epitopes being hidden by the lipoprotein matrix.

In order to evaluate these two possibilities, we used a membrane insertion assay that is independent of host serum lipoproteins. Compared to lipoprotein capture, the membrane insertion assay resulted in higher S/N in all tested TB patients, but the magnitude of the difference varied from patient to patient. In all, our results indicate clearly that an assay modality that is independent of variable host factors (membrane insertion) is more sensitive than one that is dependent on them (lipoprotein capture). The enhanced sensitivity of membrane insertion compared to lipoprotein capture is based on the optimization assays and the 8 clinical samples. The increased sensitivity of the membrane insertion assay over lipoprotein capture assay, even in spiked samples, indicates some of the LAM associated with lipoproteins is not readily detected. Indeed, a variety of factors can impact host lipoprotein concentrations, including HIV/AIDS [29, 30]. Because of insufficient volume of clinical samples, we were not able to quantitate lipoprotein concentrations to formally establish an association between serum concentrations of these host lipoproteins and LAM S/N in TB patients, and this is a weakness of our study. HIV is associated with quantitative and qualitative lipid abnormalities including low levels of HDL, disordered HDL metabolism, and reduced Apo A levels [31–33]. It is intriguing to speculate that HIV effects on host lipoproteins might influence host handling of MTB LAM, thereby impacting TB disease pathophysiology in addition to impacting performance of our lipoprotein capture assay. A better understanding of the mechanisms and kinetics of LAM sequestration and clearance could have important implications for understanding tuberculosis and inflammation more broadly.

The second unexpected finding was that serum LAM was detectable (using the lipoprotein capture assay and threshold S/N > 1.0) in over half of TB symptomatic individuals whose sputum and blood cultures all were negative for MTB. There are two possible explanations for this: 1) These are false positive results, and detected signal in the absence of LAM; or 2) MTB

LAM was present in serum, but sputum and blood cultures were falsely negative. Our existing data cannot tease apart these possibilities. However, we note that all of these individuals were enrolled with suspected TB disease, and that our assay did not have any false positive measurements in the control group (0/6). Further, the recognized sensitivity limitations of mycobacterial culture as a gold standard as well as the recognition that "TB" is nonbinary and represents a spectrum of conditions including incipient and subclinical TB, support further investigation of serum LAM as a biomarker [34, 35].

There are important limitations of our exploratory study. The sample size was small, and adult controls all were HIV-negative whereas individuals with TB symptoms all were HIV-positive. Second, as noted above, serum specimen volumes precluded performance of both LAM detection methods and HDL quantitation on all specimens, and therefore we were not able to comprehensively characterize the associations between serum LAM, host lipoproteins, and HIV serostatus. We hope to address these limitations in future clinical evaluations that are curated to address out needs.

In conclusion, we present two tailored assay strategies for the direct detection of amphiphilic serum LAM. Our findings highlight the role that host pathogen interactions play in pathogen amphiphile presentation and the need to account for these interactions in the design of diagnostic assays. Our findings also raise the intriguing possibility that serum LAM might be an informative TB biomarker of incipient or subclinical TB.

## Supporting information

**S1 File.**
(PDF)

**S1 Fig. Antibody screening and selection by colorimetric sandwich immunoassays.** Performance of antibody clones 24 (square), 27 (circle) and 31(triangle), using antibody 171 as the reporter, as assessed by sandwich colorimetric immunoassays is plotted as a function of concentration (n = 3, per antibody, per concentration). Plot shows highest absorbance of clone 31, followed by 24, and finally, 27.
(DOCX)

## Acknowledgments

We thank Mr. Aaron S. Anderson and Dr. Jessica Kubicek-Sutherland for technical guidance, assistance and helpful discussions during the course of this work. We are grateful to the patients from Uganda for their participation in the study. The authors thank Dr. Mark Perkins, who was then at the Foundation for Innovative New Diagnostics and now, at the World Health Organization, for his help in establishing the initial collaborations essential for this work. The work was performed at the Los Alamos National Laboratory, operated by Triad National Security LLC.

## Author Contributions

**Conceptualization:** Basil I. Swanson, Susan E. Dorman, Harshini Mukundan.

**Data curation:** Shailja Jakhar, Laura Lilley, Nicolas Hengartner, Emmanuel Moreau, Susan E. Dorman, Harshini Mukundan.

**Formal analysis:** Shailja Jakhar, Laura Lilley, Nicolas Hengartner, Emmanuel Moreau, Susan E. Dorman, Harshini Mukundan.

**Funding acquisition:** Susan E. Dorman, Harshini Mukundan.

**Investigation:** Shailja Jakhar, Dung Vu, Basil I. Swanson, Emmanuel Moreau, Susan E. Dorman, Harshini Mukundan.

**Methodology:** Shailja Jakhar, Ramamurthy Sakamuri, Dung Vu, Priya Dighe, Loreen R. Stromberg, Harshini Mukundan.

**Project administration:** Harshini Mukundan.

**Resources:** Basil I. Swanson, Emmanuel Moreau, Harshini Mukundan.

**Supervision:** Harshini Mukundan.

**Validation:** Shailja Jakhar.

**Visualization:** Basil I. Swanson, Harshini Mukundan.

**Writing – original draft:** Shailja Jakhar, Emmanuel Moreau, Susan E. Dorman, Harshini Mukundan.

**Writing – review & editing:** Shailja Jakhar, Ramamurthy Sakamuri, Dung Vu, Priya Dighe, Loreen R. Stromberg, Laura Lilley, Nicolas Hengartner, Basil I. Swanson, Emmanuel Moreau, Susan E. Dorman, Harshini Mukundan.

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
