## [Decision Letter · Decision Letter 0]

11 Dec 2020

PONE-D-20-35762

Interaction of Amphiphilic Lipoarabinomannan with Host Carrier Lipoproteins in Tuberculosis Patients: Implications for Blood-based Diagnostics.

PLOS ONE

Dear Dr. Mukundan,

Thank you for submitting your manuscript to PLOS ONE. After careful consideration, we feel that it has merit but does not fully meet PLOS ONE’s publication criteria as it currently stands. Therefore, we invite you to submit a revised version of the manuscript that addresses the serious concerns raised by both reviewers.

We look forward to receiving your revised manuscript.

Kind regards,

Jérôme Nigou

Academic Editor

PLOS ONE

Journal Requirements:

"I have read the journal's policy and the authors of this manuscript have the following competing interests: [insert competing interests here]"

We note that one or more of the authors are employed by a commercial company: Bako Diagnostics.

3.1. Please provide an amended Funding Statement declaring this commercial affiliation, as well as a statement regarding the Role of Funders in your study. If the funding organization did not play a role in the study design, data collection and analysis, decision to publish, or preparation of the manuscript and only provided financial support in the form of authors' salaries and/or research materials, please review your statements relating to the author contributions, and ensure you have specifically and accurately indicated the role(s) that these authors had in your study. You can update author roles in the Author Contributions section of the online submission form.

3.2. Please also provide an updated Competing Interests Statement declaring this commercial affiliation along with any other relevant declarations relating to employment, consultancy, patents, products in development, or marketed products, etc.  

5. Please include your tables only as part of your main manuscript and remove the individual files. Please note that supplementary tables should be uploaded as separate "supporting information" files.

Reviewers' comments:

Reviewer's Responses to Questions

**Comments to the Author**

1. Is the manuscript technically sound, and do the data support the conclusions?

Reviewer #1: Partly

Reviewer #2: No

2. Has the statistical analysis been performed appropriately and rigorously? 

Reviewer #1: N/A

Reviewer #2: Yes

3. Have the authors made all data underlying the findings in their manuscript fully available?

Reviewer #1: Yes

Reviewer #2: Yes

4. Is the manuscript presented in an intelligible fashion and written in standard English?

Reviewer #1: Yes

Reviewer #2: Yes

5. Review Comments to the Author

Reviewer #1: The study presents two methods for detection of lipoarabinomannan (LAM) in serum, as a tool to diagnose tuberculosis. Two methods were evaluated, the lipoprotein capture assay and the membrane insertion assay, both intended to solve the important problem with interaction of the amphiphilic LAM with host carrier lipoproteins. A very small number of serum samples are used to evaluate the two methods. It would be valuable to know more about the individual samples, and also about the possible use of these methods in clinical practice.

The patient samples were selected from existing stored specimens that previously had been obtained for a study that evaluated the diagnostic accuracy of the Alere Determine™ TB LAM Ag assay (ref 27). In that study 1013 HIV-positive participants were enrolled. Among culture-positive tuberculosis patients, the Alere test identified 136/367 (37.1%) overall and 116/196 (59.2%) in the group with CD4 ≤100 cells per cubic millimeter.

From these specimens serum from 35 verified HIV positive TB cases, and 13 individuals with TB symptoms but no positive cultures, and 6 HIV negative controls were selected for this study. It would be important to know 1) the CD4 status of the 35 culture-confirmed TB patients and the 13 individuals with TB symptoms that were selected for inclusion in this study. The fact that the 6 control samples were from HIV-negative individuals while the 13 individuals with TB symptoms were HIV positive makes this information even more important.

It would also be important to know whether the samples were selected with knowledge of their Alere results.

The samples were too few to evaluate the sensitivity and specificity of the tests, especially the membrane insertion assay: only 8 samples were used to evaluate the membrane insertion assay, in parallel with the lipoprotein capture assay, and there were only 6 control samples, all negative in both assays. Based on the comparison between the two assays in the 8 samples the authors conclude that the membrane insertion assay was the most sensitive one.

It would also be important to know how the authors envisage how the methods could be developed for use in clinical practice, since they include sophisticated instruments in both methods as well as harsh treatments (chloroform:methanol extraction) in the membrane insertion assay. The need to release LAM from host lipoprotein matrix is at the core of most work on development of methods to detect and quantify LAM in patient fluids, and crude chloroform:methanol extraction poses several problems.

From this point of view the lipoprotein capture assay seems more interesting, albeit apparently less effective, probably due to LAM epitopes being hidden by the lipoprotein matrix, a fact that should be discussed.

Minor comment: Reference 17 is redundant and should be excluded or replaced by more recent references.

Reviewer #2: The manuscript “Interaction of Amphiphilic Lipoarabinomannan with Host Carrier Lipoproteins in Tuberculosis Patients: Implications for Blood-based Diagnostics” by Mukundan et al. proposes two novel sandwich immunoassays for the detection of lipoarabinomannan derived from Mycobacterium tuberculosis in the blood. The assays are comprised by a capture element and a labeling element. The capture elements leverage LAM association with lipid particles or lipophilic phase and encompass: 1) lipid bilayer and 2) anti high-density lipoprotein antibody associated to a lipid bilayer. Labeling element uses an anti-LAM antibody, clone CS40.

The paper is well written and clearly presented. It has intellectual merit because it aims to bring additional evidence that LAM exists in the blood of tuberculosis patients in association with lipoproteins. The labeling antibody CS-40 mAb is well characterized in the literature and showed epitope specificity (DOI: https://doi.org/10.4049/jimmunol.1701673). Data present in the supplementary information support the choice of this antibody in comparison to other clones.

However, the design of the assay lacks rigor. The main issue is that “specificity controls were performed using IgG labelled with AF-647, rather than anti-LAM antibodies”. In order to ensure valid readings, immunoassay controls must be run using all the reagents used for the unknown samples (https://www.ncbi.nlm.nih.gov/books/NBK92434/). In the proposed assay it is especially crucial that the anti-LAM antibody is used for specificity controls, because it is the only element of the assay that confers specificity. Furthermore, the epitope of the IgG used for the negative control readings is not specified. On a minor note, the advantages of using an anti-HDL antibody linked to a lipid bilayer versus a less complicate alternative, such as ELISA plate coating, are not elaborated.

Limit of detection is usually presented as a general characteristic of the assay and not in function of the concentration of one sample (DOI: https://doi.org/10.4049/jimmunol.1701673). In the results section, the limit of detection of assay number one is not presented, and the limit of detection of assay number two is 8.5 nM, which for LAM corresponds roughly to 400 ng/mL. Discussion on the expected concentration range of LAM in serum is lacking. Llimit Since S/N values are reported for patients, S/N curves for different concentration of spiked control serum samples should be presented (Fig. 2b, 3b). Figure 5b reports discrepant values obtained from the two assays in individual samples, but a strategy to reconcile the values to obtain a quantitative assay is missing.

The abstract should indicate that the assays are sandwich assays and use anti-LAM CS-40 mAb as a labeling antibody.

Demographic and clinical information of patients should be included in the manuscript if available.

6. PLOS authors have the option to publish the peer review history of their article (what does this mean?). If published, this will include your full peer review and any attached files.

Reviewer #1: **Yes: **Gunilla Källenius

Reviewer #2: No

---

## [Author Response · Author response to Decision Letter 0]

28 Jan 2021

Please find attached detailed responses to reviewers 1 and 2. These responses are also included in the attached "Response to Reviewers" document, along with the editorial questions raised. Also, the tracked version of the manuscript reflects these changes as embedded in the manuscript. 

Reviewer #1: 

1. The study presents two methods for detection of lipoarabinomannan (LAM) in serum, as a tool to diagnose tuberculosis. Two methods were evaluated, the lipoprotein capture assay and the membrane insertion assay, both intended to solve the important problem with interaction of the amphiphilic LAM with host carrier lipoproteins. A very small number of serum samples are used to evaluate the two methods. It would be valuable to know more about the individual samples, and also about the possible use of these methods in clinical practice.

Response: We have now included a table (Table S2) which captures more information on the samples in the revised manuscript. Specifically, gender, age, CD4 cells/mm3, identification of infiltrates and miliary infiltrates, cavity and urine LAM diagnostics (where available) have been included.

Patient ID Sex Age (years) CD4 cells/mm3 Infiltrates Miliary infiltrates Cavity Urine LAM result

1 FEMALE 28 32 1

2 FEMALE 58 57 YES NO NO 1

3 FEMALE 39 1 1

4 MALE 31 5 YES NO NO 1

5 FEMALE 38 202 1

6 MALE 35 125 YES NO NO 1

7 MALE 35 32 1

8 FEMALE 25 25 YES YES NO 1

9 MALE 30 18 YES NO NO 1

10 MALE 45 230 YES NO NO 1

11 MALE 37 2 1

12 FEMALE 35 313 YES NO NO 0

13 FEMALE 27 59 YES NO NO 1

14 FEMALE 29 10 YES YES NO 1

15 MALE 43 10 YES NO NO 1

16 FEMALE 21 138 YES NO NO 1

17 FEMALE 38 27 1

18 MALE 25 14 YES NO NO 1

19 FEMALE 38 28 YES YES NO 1

20 MALE 30 61 YES YES NO 1

21 FEMALE 30 31 YES YES NO 0

22 FEMALE . 305 YES NO YES 0

23 MALE 28 109 YES NO NO 0

24 FEMALE 21 199 YES NO NO 0

25 MALE 50 2 YES NO NO 0

26 FEMALE 32 348 YES NO NO 0

27 FEMALE 28 117 YES NO NO 0

28 FEMALE 34 7 0

29 FEMALE 29 616 YES NO YES 0

30 FEMALE 18 13 YES NO NO 0

31 FEMALE 48 56 YES YES NO 0

32 FEMALE 24 81 YES NO NO 0

33 MALE 33 14 YES NO NO 0

34 FEMALE 24 354 0

35 MALE 50 54 YES NO NO 0

36 FEMALE 37 67 YES NO NO 1

37 FEMALE 40 25 1

38 MALE 44 36 YES YES NO 0

39 FEMALE 29 . 

40 MALE . 626 YES NO NO 1

41 FEMALE 19 19 0

42 FEMALE 29 635 YES NO NO 0

43 FEMALE 20 73 0

44 MALE . 28 YES NO NO 0

45 MALE . 206 NO NO NO 0

46 FEMALE 30 138 YES NO NO 0

47 FEMALE 28 36 YES NO NO 0

48 FEMALE 33 140 YES NO NO 0

Additionally, we used existing de-identified serum specimens from six Ugandan adults who did not have TB symptoms and were not known to be HIV-positive; no additional meta-data were available for these specimens. Included in manuscript on page 9, lines 160-163.

2. The patient samples were selected from existing stored specimens that previously had been obtained for a study that evaluated the diagnostic accuracy of the Alere Determine™ TB LAM Ag assay (ref 27). In that study 1013 HIV-positive participants were enrolled. Among culture-positive tuberculosis patients, the Alere test identified 136/367 (37.1%) overall and 116/196 (59.2%) in the group with CD4 ≤100 cells per cubic millimeter.

From these specimens, serum from 35 verified HIV positive TB cases, and 13 individuals with TB symptoms but no positive cultures, and 6 HIV negative controls were selected for this study. It would be important to know 1) the CD4 status of the 35 culture-confirmed TB patients and the 13 individuals with TB symptoms that were selected for inclusion in this study. The fact that the 6 control samples were from HIV-negative individuals while the 13 individuals with TB symptoms were HIV positive makes this information even more important

Response: The CD4 status of the patients has now been included in Table S2 of the revised supplemental document and a discussion of this has been included on Page 9, Lines 160-163 of the revised manuscript. 

3. It would also be important to know whether the samples were selected with knowledge of their Alere results. 

Response: Samples were selected on the basis of culture results, and sent to LANL in a double blinded fashion. The Alere results were not taken into consideration for sample selection. Included on page 9, lines 159-160 of revised manuscript in clinical specimens section.

4. The samples were too few to evaluate the sensitivity and specificity of the tests, especially the membrane insertion assay: only 8 samples were used to evaluate the membrane insertion assay, in parallel with the lipoprotein capture assay, and there were only 6 control samples, all negative in both assays. Based on the comparison between the two assays in the 8 samples the authors conclude that the membrane insertion assay was the most sensitive one.

Response: We agree with the reviewer that a much larger cohort should be analyzed to determine performance metrics in the clinical samples. This study was focused on establishing and optimizing the methods for more sensitive detection of serum LAM, and validating efficacy in clinical samples. To this end, the optimization of the study for both membrane insertion and lipoprotein capture (Figure 2b and Figure 3b) was performed in serum spiked with varying concentrations of LAM. The enhanced sensitivity of the membrane insertion assay over lipoprotein capture is noted in these spiked samples for known concentrations of the antigen. This trend is replicated in the small subset of clinical samples that we evaluated. This has been clarified in page 24, lines 481-486 of the revised manuscript. 

It would also be important to know how the authors envisage how the methods could be developed for use in clinical practice, since they include sophisticated instruments in both methods as well as harsh treatments (chloroform:methanol extraction) in the membrane insertion assay. The need to release LAM from host lipoprotein matrix is at the core of most work on development of methods to detect and quantify LAM in patient fluids, and crude chloroform:methanol extraction poses several problems.

Response: Our team has been working on resolving the engineering and simplification challenges for use of this technology at the point of need. We have developed and validated the first microfluidics device which is capable of performing lipid extractions within minutes at the point of need (Manuscript accepted with minor revisions pending to Nature scientific reports, U.S. Provisional Patent Application No. 63/113,310). In addition, we have simplified the waveguide-based optical biosensor for field deployment, and validated assay performance on this new portable (<10Lbs) instrument with optical components integrated and the software for functionality on a phone-based application. This manuscript is in preparation, provisional patent application No S133706.000, Entitled “Microfluidics Device”). We have added a generic description of these developments in lines 331-337, page 17 of the revised manuscript. 

5. From this point of view the lipoprotein capture assay seems more interesting, albeit apparently less effective, probably due to LAM epitopes being hidden by the lipoprotein matrix, a fact that should be discussed.

Response: We agree with the reviewer that both of these assays have relatively different advantages and disadvantages. With the developments in progress, as outlined in response to the previous question, we anticipate the membrane insertion assay will also become more field friendly. However, for the purpose of this manuscript, we have included a more detailed description of the advantages and disadvantages of the two methods, while briefly mentioning the developments in progress. These changes have now been included on page 15, lines 303-304, page 17, lines 331-337, page 6- lines 99-104 of the revised manuscript. 

The increased sensitivity of the membrane insertion assay over lipoprotein capture assay, even in spiked samples, indicates some of the LAM associated with lipoproteins is not readily detected. We agree with this assessment and have clarified this in the discussion on page 24, lines 486-488 of the revised manuscript. 

6. Minor comment: Reference 17 is redundant and should be excluded or replaced by more recent references. 

Response: This reference has been removed in the revised manuscript. 

 

Reviewer #2: The manuscript “Interaction of Amphiphilic Lipoarabinomannan with Host Carrier Lipoproteins in Tuberculosis Patients: Implications for Blood-based Diagnostics” by Mukundan et al. proposes two novel sandwich immunoassays for the detection of lipoarabinomannan derived from Mycobacterium tuberculosis in the blood. The assays are comprised by a capture element and a labeling element. The capture elements leverage LAM association with lipid particles or lipophilic phase and encompass: 1) lipid bilayer and 2) anti high-density lipoprotein antibody associated to a lipid bilayer. Labeling element uses an anti-LAM antibody, clone CS40.

The paper is well written and clearly presented. It has intellectual merit because it aims to bring additional evidence that LAM exists in the blood of tuberculosis patients in association with lipoproteins. The labeling antibody CS-40 mAb is well characterized in the literature and showed epitope specificity (DOI: https://doi.org/10.4049/jimmunol.1701673). Data present in the supplementary information support the choice of this antibody in comparison to other clones.

1. However, the design of the assay lacks rigor. The main issue is that “specificity controls were performed using IgG labelled with AF-647, rather than anti-LAM antibodies”. In order to ensure valid readings, immunoassay controls must be run using all the reagents used for the unknown samples (https://www.ncbi.nlm.nih.gov/books/NBK92434/). In the proposed assay it is especially crucial that the anti-LAM antibody is used for specificity controls, because it is the only element of the assay that confers specificity. Furthermore, the epitope of the IgG used for the negative control readings is not specified.

Response: We apologize if our original manuscript was not clear on the methods used. 

a) Whereas CS40, anti-LAM antibody procured from BEI, was also evaluated, our initial assessments revealed that clones 171 and 24 from FIND performed best in our assay, as indicated in supplemental figure S1. Thus, all the developments described in the manuscript and the clinical evaluations were performed using a combination of FIND clones 171 and 24. 

b) All specificity controls were also performed using FIND 171 and 24 as shown in supplemental figure S1, and not using a generic IgG labeled with AF-647. Thus, every single non-specific measurement reported in the manuscript was performed using the exact combination of anti-LAM antibodies as the specific measurements, and under exactly the same conditions of the assay. We have clarified these points in the methods section of the revised manuscript, on page 14, lines 280-281. 

2. On a minor note, the advantages of using an anti-HDL antibody linked to a lipid bilayer versus a less complicate alternative, such as ELISA plate coating, are not elaborated. 

Response: We have elaborated on the advantages of the lipoprotein capture and membrane insertion assays, as compared to a conventional plate-based ELISA more clearly in the revised manuscript on page 7, lines 108-121. There are two limitations that impact the use of conventional plate-based methods for the detection of LAM. 1) LAM is an amphiphile and therefore, sticks to plastics and other surfaces, reducing assay reliability, sensitivity and reproducibility. 2) In clinical samples, LAM is already sequestered by host lipoproteins and not available in free-form(1). It has to be detected in this conformation or liberated from it to facilitate detection. In this manuscript, we have attempted to develop methods that address both these limitations. Both the assays are performed on a surface functionalized with lipid bilayer, and therefore override the challenges of using plastic ELISA plates in concert with amphiphiles. Further, we have consistently shown that the use of the waveguide platform offers at least 10X greater sensitivity than conventional plate-based ELISA, which is an advantage when it comes to sensitive detection of pathogen antigens in complex clinical samples(2,3).

3. Limit of detection is usually presented as a general characteristic of the assay and not in function of the concentration of one sample (DOI: https://doi.org/10.4049/jimmunol.1701673). In the results section, the limit of detection of assay number one is not presented, and the limit of detection of assay number two is 8.5 nM, which for LAM corresponds roughly to 400 ng/mL. Discussion on the expected concentration range of LAM in serum is lacking. Limit Since S/N values are reported for patients, S/N curves for different concentration of spiked control serum samples should be presented (Fig. 2b, 3b). 

Response: We have clarified the quantification of the assays more rigorously in the revised manuscript. 

The concentration of LAM in patient serum has not been clearly established. Brock et al. showed a range of 0-132 pg/ml in individuals without HIV infection(4). Other studies have “extracted” LAM from serum, but direct measurement of the antigen has not been largely reported(5–7). 

According to equation 2, LOD for lipoprotein capture assay is 69 nM, which corresponds to a concentration of 1900ng/mL of LAM. LOD for membrane insertion assay is 3.3 nM, which corresponds to a concentration of 114 ng/mL LAM. This clearly demonstrates the greater sensitivity of the membrane insertion assay over the lipoprotein capture assay in standard curves based on spiked samples. At 500 nM of LAM, the S/N by lipoprotein capture assay is 8.2, whereas that by membrane insertion assay is 782. 

Because of the significant difference in fluorescence units between lipoprotein capture and membrane insertion for some of the patients, we have chosen to present the data in terms of S/N. However, we have included the conversion metrics above clearly in the revised manuscript to ensure the interpretation of the data is clear to the reviewer and the reader. 

4. Figure 5b reports discrepant values obtained from the two assays in individual samples, but a strategy to reconcile the values to obtain a quantitative assay is missing.

The abstract should indicate that the assays are sandwich assays and use anti-LAM CS-40 mAb as a labeling antibody.

Response: Figure 5b represents the difference in sensitivity for the same clinical sample when two different methods are used to measure the antigen. One method is more sensitive than the other, and is responsible for the disparity in the signature intensity noted. We have expanded on this, and clarified the results, in the results section more clearly (page 22, line 440-442). 

The antibody combination used in the assay was FIND Clones 171 and 24, and this has been clarified in the legends and in the methods as well. 

5. Demographic and clinical information of patients should be included in the manuscript if available.

Response: We have now included Table S2 in the revised manuscript, which includes meta-data with this information for the patients in the study. 

References:

1. Sakamuri RM, Price DN, Lee M, Cho SN, Barry CE, Via LE, et al. Association of lipoarabinomannan with high density lipoprotein in blood: Implications for diagnostics. Tuberculosis. 2013;93(3):301–7. 

2. Mukundan H, Kubicek JZ, Holt A, Shively JE, Martinez JS, Grace K, et al. Planar optical waveguide-based biosensor for the quantitative detection of tumor markers. Sensors Actuators, B Chem. 2009;138(2):453–60. 

3. Kubicek-Sutherland JZ, Vu DM, Mendez HM, Jakhar S, Mukundan H. Detection of lipid and amphiphilic biomarkers for disease diagnostics. Biosensors. 2017;7(3):25. 

4. Brock M, Hanlon D, Zhao M, Pollock NR. Detection of mycobacterial lipoarabinomannan in serum for diagnosis of active tuberculosis. Diagn Microbiol Infect Dis. 2020; 

5. Sada E, Aguilar D, Torres M, Herrera T. Detection of lipoarabinomannan as a diagnostic test for tuberculosis. J Clin Microbiol. 1992;30(9):2415–2418. 

6. Crawford AC, Laurentius LB, Mulvihill TS, Granger JH, Spencer JS, Chatterjee D, et al. Detection of the tuberculosis antigenic marker mannose-capped lipoarabinomannan in pretreated serum by surface-enhanced Raman scattering. Analyst [Internet]. 2017;142(1):186–96. Available from: http://xlink.rsc.org/?DOI=C6AN02110G

7. Owens NA, Young CC, Laurentius LB, De P, Chatterjee D, Porter MD. Detection of the tuberculosis biomarker mannose-capped lipoarabinomannan in human serum: Impact of sample pretreatment with perchloric acid. Anal Chim Acta. 2019;1046:140–7.

---

## [Decision Letter · Decision Letter 1]

2 Mar 2021

Interaction of Amphiphilic Lipoarabinomannan with Host Carrier Lipoproteins in Tuberculosis Patients: Implications for Blood-based Diagnostics.

PONE-D-20-35762R1

Dear Dr. Mukundan,

We’re pleased to inform you that your manuscript has been judged scientifically suitable for publication and will be formally accepted for publication once it meets all outstanding technical requirements. Please incorporate in the text the minor modifications suggested by the reviewers.

Kind regards,

Jérôme Nigou

Academic Editor

PLOS ONE

Additional Editor Comments (optional):

Reviewers' comments:

Reviewer's Responses to Questions

**Comments to the Author**

1. If the authors have adequately addressed your comments raised in a previous round of review and you feel that this manuscript is now acceptable for publication, you may indicate that here to bypass the “Comments to the Author” section, enter your conflict of interest statement in the “Confidential to Editor” section, and submit your "Accept" recommendation.

Reviewer #1: (No Response)

Reviewer #2: All comments have been addressed

2. Is the manuscript technically sound, and do the data support the conclusions?

Reviewer #1: Yes

Reviewer #2: Yes

3. Has the statistical analysis been performed appropriately and rigorously? 

Reviewer #1: N/A

Reviewer #2: N/A

4. Have the authors made all data underlying the findings in their manuscript fully available?

Reviewer #1: Yes

Reviewer #2: Yes

5. Is the manuscript presented in an intelligible fashion and written in standard English?

Reviewer #1: Yes

Reviewer #2: Yes

6. Review Comments to the Author

Reviewer #1: 1. As pointed out in the previous review It would also be important to know how the authors envisage how the methods could be developed for use in clinical practice. The authors have now added information on unpublished work on how that has been done on line 331-337. However these sentences do not belong there (in M6M). They should be in the Discussion, with reference to how they foresee a more userfriendly test, indicating (unpublished results or similar.

2. An additional comment: on line 505-506 the authors write "2) MTB LAM was present in serum, but sputum and blood cultures were falsely negative". I would not say "falsely". They were negative but the patients may still have eg extrapulmonary TB.

Reviewer #2: The authors have addressed all the questions that were raised.

However, I believe it would be very helpful to readers if the discussion about expected LAM concentrations in serum provided by the authors in the response to reviewers' section is incorporated in the discussion section of the paper. Answer: "The concentration of LAM in patient serum has not been clearly established. Brock et al. showed a range of 0-132 pg/ml in individuals without HIV infection(4). Other studies have “extracted” LAM from serum, but direct measurement of the antigen has not been largely reported(5–7)." This literature review and a short discussion on the apparent discrepancy between LAM concentrations detected in this paper and the suggested concentration range suggested by Brock et al., (4), would provide helpful information on how the present study relates to the state of the field.

7. PLOS authors have the option to publish the peer review history of their article (what does this mean?). If published, this will include your full peer review and any attached files.

Reviewer #1: **Yes: **Gunilla Källenius

Reviewer #2: No

---

## [Editor Report · Acceptance letter]

15 Mar 2021

PONE-D-20-35762R1 

Interaction of amphiphilic lipoarabinomannan with host carrier lipoproteins in tuberculosis patients: Implications for blood-based diagnostics.  

Dear Dr. Mukundan:

I'm pleased to inform you that your manuscript has been deemed suitable for publication in PLOS ONE. Congratulations! Your manuscript is now with our production department. 

Kind regards, 

on behalf of

Dr. Jérôme Nigou 

Academic Editor

PLOS ONE